# The Spliceosome: A New Therapeutic Target in Chronic Myeloid Leukaemia

**DOI:** 10.3390/cancers14194695

**Published:** 2022-09-27

**Authors:** Benjamin Lebecque, Celine Bourgne, Chinmay Munje, Juliette Berger, Thomas Tassin, Pascale Cony-Makhoul, Agnès Guerci-Bresler, Hyacinthe Johnson-Ansah, Wei Liu, Sandrine Saugues, Andrei Tchirkov, David Vetrie, Mhairi Copland, Marc G. Berger

**Affiliations:** 1Hématologie Biologique, CHU Estaing, 63000 Clermont-Ferrand, France; 2Equipe d’Accueil 7453 CHELTER, Université Clermont Auvergne, 63001 Clermont-Ferrand, France; 3Paul O’Gorman Leukaemia Research Centre, Institute of Cancer Sciences, University of Glasgow, Glasgow G12 8QQ, UK; 4CH Annecy-Genevois, 74374 Pringy, France; 5Groupe Fi-LMC, Centre Léon Bérard, 69008 Lyon, France; 6Hématologie Clinique, CHRU Brabois, 54500 Vandoeuvre-lès-Nancy, France; 7Institut d’Hématologie de Basse Normandie, CHU, 14033 Caen, France; 8Wolfson Wohl Cancer Research Centre, Institute of Cancer Sciences, University of Glasgow, Glasgow G12 8QQ, UK; 9Cytogénétique Médicale, CHU Clermont-Ferrand, CHU Estaing, 63000 Clermont-Ferrand, France

**Keywords:** CP-CML, spliceosome, CD34^+^ cells

## Abstract

**Simple Summary:**

RNA splicing factors are frequently altered in cancer and have been found mutated or deregulated in myeloid malignancies, justifying the growing interest in new therapeutic strategies. We recently showed that the DNA methylation alterations of CD34^+^CD15^−^ chronic myeloid leukaemia (CML) cells affect alternative splicing genes, suggesting that spliceosome actors might be altered in chronic-phase (CP)-CML. We investigated the expression of 12 splicing genes in primary CP-CML CD34^+^ cells at diagnosis (*n* = 15). We found that CP-CML CD34^+^ cells had a distinct splicing signature profile, suggesting: (i) a spliceosome deregulation from the diagnosis time and (ii) an intraclonal heterogeneity. In vitro incubation of a spliceosome-targeted drug (TG003) showed that CP-CML CD34^+^ cells are spliceosome dependent; moreover, with the combination of TKI, the two drugs showing an additive effect while sparing healthy donors cells. Our results suggest that the spliceosome may be a new potential target for the treatment of CML.

**Abstract:**

RNA splicing factors are frequently altered in cancer and can act as both oncoproteins and tumour suppressors. They have been found mutated or deregulated, justifying the growing interest in the targeting of splicing catalysis, splicing regulatory proteins, and/or specific, key altered splicing events. We recently showed that the DNA methylation alterations of CD34^+^CD15^−^ chronic myeloid leukaemia (CML) cells affect, among others, alternative splicing genes, suggesting that spliceosome actors might be altered in chronic-phase (CP)-CML. We investigated the expression of 12 spliceosome genes known to be oncogenes or tumour suppressor genes in primary CP-CML CD34^+^ cells at diagnosis (*n* = 15). We found that CP-CML CD34^+^ cells had a distinct splicing signature profile as compared with healthy donor CD34^+^ cells or whole CP-CML cells, suggesting: (i) a spliceosome deregulation from the diagnosis time and (ii) an intraclonal heterogeneity. We could identify three profile types, but there was no relationship with a patient’s characteristics. By incubating cells with TKI and/or a spliceosome-targeted drug (TG003), we showed that CP-CML CD34^+^ cells are both BCR::ABL and spliceosome dependent, with the combination of the two drugs showing an additive effect while sparing healthy donors cells. Our results suggest that the spliceosome may be a new potential target for the treatment of CML.

## 1. Introduction

The spliceosome is a macromolecular complex that generates distinct mRNA isoforms from a single pre-mRNA using specific sequences (donor and acceptor sites) present in introns. In haematopoiesis, the spliceosome plays a major role in regulating cell differentiation, particularly in the erythroid lineage [1], and is essential for progenitor/stem cell status regulation [2]. Recently, it has been shown that the spliceosome complex is frequently altered in cancer, mainly due to the occurrence of mutations in genes encoding spliceosome proteins and/or due to the deregulated expression of genes with key roles in splicing [3]. For example, splicing factor 3b subunit 1 (*SF3B1*) mutations have been described in myelodysplastic syndrome (MDS) [4], and deregulated heterogeneous nuclear ribonucleoprotein A1 (hnRNPA1) expression has occurred in solid cancers [3]. Moreover, accumulating evidence indicates that several cancer molecular subtypes are highly dependent on splicing for cell survival [5]. These findings have resulted in a growing interest in the therapeutic targeting of splicing enzymes, splicing regulatory proteins, and/or key splicing events that are altered in cancer. Different classes of small molecules that target different elements of the splicing machinery are available for in vitro use, such as pladienolides, herboxidienes, spliceostatins, CDC-like kinase (CLK) inhibitors, and SPRK inhibitors (reviewed in Lee S. et al. [6]). Some of these agents are currently being assessed in clinical trials (e.g., the *SF3B1* inhibitor H3B-8800 in MDS, and the CLK inhibitor SM08502in solid tumours) [7].

Chronic myeloid leukaemia (CML) is an interesting model for spliceosome studies. Indeed, CML is a clonal disorder which develops from a single haematopoietic stem cell (HSC) that acquires the Philadelphia chromosome, a unique reciprocal translocation, t(9;22)(q34;q11), between the long arms of chromosome 9 and chromosome 22 [8]. This event results in the expression of a chimeric oncoprotein, termed BCR::ABL, with constitutively active tyrosine kinase activity. Tyrosine kinase inhibitors (TKIs) against BCR::ABL are a model example of targeted therapy, and have revolutionized CML treatment and prognosis. Currently, CML management consists of long-term TKI therapy with regular monitoring of *BCR::ABL1* by qRT-PCR for the detection of residual disease.

Emerging evidence indicates that CML is sustained by a small pool of leukaemic stem cells (LSCs) that have a self-renewing capacity and reside within the bone marrow niche. A subgroup of these primitive LSCs are insensitive to TKIs (e.g., imatinib, dasatinib, nilotinib) and are, therefore, difficult to eradicate. Multiple mechanisms, including both BCR::ABL-dependent and -independent mechanisms, have been described, but they only partially explain the TKI resistance of LSCs [9]. Consequently, the features of the immature cell subset within the CP-CML cells must be thoroughly investigated to identify new therapeutic strategies that could be combined with TKIs to deepen responses and improve outcomes for CML patients. We recently identified a specific DNA methylation profile of chronic-phase CML (CP-CML) at diagnosis, and within the CP-CML cells, abnormalities that are specific to immature CD34^+^CD15^−^ progenitor cells were found, particularly in genes subject to alternative splicing [10]. This suggests that alternative splicing regulation might be altered in CML. Data implicating the spliceosome in TKI resistance mechanisms in CML are very limited. Spliceosome gene mutations have been rarely described in CML [11], unlike in other myeloid neoplasms (acute myeloid leukaemia, MDS, myelodysplastic/myeloproliferative neoplasms) [4,12]. Few studies have assessed the expression of splicing genes in CML cells, and none the effect of drugs targeting the spliceosome complex in CD34^+^ CML cells. In this study, we investigated the expression of twelve splicing genes in CD34^+^CD15^−^ CP-CML samples and identified three distinct splicing signatures at diagnosis. Moreover, we tested three spliceosome-targeted drugs in CD34^+^ CP-CML cells. Our results suggest that the spliceosome could be a critical part of the CML progenitor cell machinery and a potential therapeutic target.

## 2. Materials and Methods

### 2.1. Cell Lines

The KCL22 and K562 cell lines are widely used in in vitro models of human leukaemia [13]. KCL22 and K562 cell lines were commercially sourced from ECACC and the German Collection of Microorganisms and Cell Cultures (DMSZ). The parental human BCR::ABL1-positive KCL22 cell line [13], derived from the pleural effusion of a 32-year-old woman with CML at the blast phase (BP-CML), has a hyperdiploid karyotype with 3.3% polyploidy (DSMZ # ACC 519). The basophilic, erythroblastic hypotriploid K562 cell line [14] was derived from the pleural effusion of a 53-year-old woman with CML at the BP-CML phase who harboured genomic mutations in *TP53* and *CDKN2A*, according to the Sanger COSMIC database (DSMZ # ACC 10). All cell line samples were locally cryopreserved in the university hospital biobank (CRB-Auvergne, certified according to the ISO 9001 and NF S 96-900 standard).

### 2.2. Cell Culture

Cell lines (KCL22, K562) were cultured in RPMI1640 supplemented with 10% foetal bovine serum, 50 U/mL of penicillin, 50 mg/mL of streptomycin, and 2 mM of L-glutamine (Invitrogen) at 37 °C, 5% CO_2_. Primary CML cells were cultured in Iscove modified Dulbecco medium (IMDM, Sigma) supplemented with a serum substitute (bovine serum albumin, insulin, transferrin; StemCell Technologies, Vancouver, BC, Canada), 0.1 mM of 2-mercaptoethanol (Sigma), L-glutamine (2 mM), penicillin/streptomycin (100 UmL^−1^), and low-density lipoprotein (0.04 mg/mL). The medium was also supplemented with a cocktail of five growth factors: 5 ng/mL of FLT3 ligand, 5 ng/mL of stem cell factor, and 1 ng/mL/each of interleukin (IL)–3, IL-6 (all from StemCell Technologies, Vancouver, BC, Canada), and granulocyte-colony stimulating factor (G-CSF; Chugai Pharma Europe, London, UK).

### 2.3. Primary Human Cell Samples

CML primary cells from peripheral blood or bone marrow were obtained from newly diagnosed patients with CP-CML after informed consent, according to the Declaration of Helsinki, in the context of two biological collections that have obtained all the legal and regulatory authorisations: IMIC and EPIK. Patients’ characteristics (Appendix A), and clinical and biological data were obtained from a multicentric database (CML Observatory) where data were centralized after obtaining informed consent from each patient.

Primary peripheral blood leukocytes were isolated from blood samples or bone marrow collected in EDTA tubes at diagnosis (for patients with CP-CML) by 155 mM ammonium chloride (NH4Cl)-based erythrocyte lysis (StemCell Technologies, Vancouver, BC, Canada). All the flow cytometry sorting experiments were carried out with fresh cells, within 24 h of sampling. For this study, the normal control peripheral blood samples were the leftover cells from samples used to evaluate the quality of G-CSF-mobilized peripheral blood progenitor cells collected by apheresis from healthy donors (HD = 10). These samples could be used for research because donors were informed and did not verbally express any objections, as stipulated by the French law.

### 2.4. Reagent

SRPIN340, TG003, and EPZ015666 were purchased from Selleckchem. All drugs were reconstituted in DMSO and stored at –80 °C as 50 mM (SRPIN340, EPZ015666) and 10 mM (TG003) stock solutions.

TG003 is a benzothiazole compound that inhibits the kinase activity of Cdc2-like kinase (Clks). CLK1 is alternatively spliced in various cancer cell lines including K562 [15]. By this intermediate, TG003 affects the phosphorylation of serine/arginine-rich proteins (SR proteins) that mediate the regulation of alternative splicing in vitro and in vivo and blocks the dissociation of nuclear speckles [16]. This drug already has different applications; TG003 was used in cancer models [17,18], in genetic diseases such as Duchenne muscular dystrophy [19], or in influenza A virus replication [20]. SRPIN340 is an ATP-competitive inhibitor of serine/arginine-rich protein kinases 1 and 2 (SRPKs) [21]. SRPKs are known to be upregulated in CML [22]. This compound affects the phosphorylation of SR family members (*SRSF1*, *SRSF2*, *SRSF4*, *SRSF5* and *SRSF6*) and modulate the expression of genes involved in cell proliferation and survival (*MAP2K1*, *MAP2K2*, *VEGF* and *FAS*). It has been used as an antiviral agent [23] and as an anti-tumour agent [24,25,26]. EPZ015666 is a *PRMT5* inhibitor. This drug has been used in different cancers, such as lymphoma [27]. *PRMT5* is a key regulator of the core splicing machinery, and is known to be upregulated in CML [28].

Imatinib and nilotinib (Sequoia Research Product, Pangbourne, UK) were dissolved in sterile DMSO. Stock solutions were prepared at 10 mM, aliquoted, and kept at −20 °C until use. All molecules were diluted in the appropriate culture medium before use to have the same percentage of DMSO in each condition.

### 2.5. Cell Viability Assays

K562 and KCL22 cells (3000 cells/well) were seeded in 96-well plates. Each well contained 100 μL of complete RPMI medium and 100 μL of drug solution at different concentrations. After 72 h of culture, resazurin (50 µM, Sigma-Aldrich, Darmstadt, Germany) was added to the wells (2 h, 37 °C), and the plate was read at 535_ex_ and 590_em_ in a microplate reader (Sinergy HT, Biotek). Each experimental procedure was performed in triplicate.

### 2.6. Cell Counting, Apoptosis Assays and Cell Cycle Analysis

Cells lines and CD34^+^ CP-CML cells were seeded at 200,000 cells/mL before the addition of drugs for 72 h. Cell viability was assessed by trypan blue exclusion (Sigma-Aldrich, Glasgow, Scotland). Apoptosis and the cell cycle were quantified using a FACSCanto II flow cytometer (BD Biosciences) after staining with Annexin-V–Allophycocyanin (APC)/7-AAD and with propidium iodide (BD Pharmingen, San Diego, CA, USA), respectively.

### 2.7. Cell Division Monitoring by Staining with Carboxyfluorescein Diacetate Succinimidyl Diester

CD34^+^ CP-CML cells were incubated with 1 μM of carboxyfluorescein diacetate succinimidyl diester (CFSE, Invitrogen, Carlsbad, CA, USA) for 72 h and stained with the anti-CD34-PerCP-A antibody (BD PharMingen, San Diego, CA, USA), Annexin-V–APC, and 7-AAD before analysis using a FACSCanto II flow cytometer (BD Biosciences, San Diego, CA, USA). Cells cultured in colcemid (Invitrogen, 100 ng/mL), which arrests cell cycle progression, were used to establish the CFSE^max^ quiescent cell population at all time points studied.

### 2.8. Colony Forming Cell (CFC) Assays

CD34^+^ cells from the peripheral blood of patients with CP-CML at diagnosis were quantified by flow cytometry after sampling. CD34^+^ cells were seeded at two cell concentrations (low: ~150 CD34^+^; high: ~300 CD34^+^) using the Methocult H84434 (STEMCELL Technologies) with or without pre-incubation with imatinib for 48 h, and then incubated with TG003 added into the culture medium for 14 days. Colonies were counted 14 days after plating.

### 2.9. Gene Expression

Total RNA was extracted using Direct-zol (Ozyme, Courtaboeuf, France) and reverse transcribed using SuperScript III (SuperScript^®^ IV First-Strand Synthesis, Thermo Fisher), according to the manufacturers’ instructions.

The standard Sybr Green real-time quantitative polymerase chain reaction (qPCR) protocol was used to assess the expression of splicing genes. *GUSB* and *SDHA* were used as housekeeping genes. Probes were designed using the sequences in the National Center for Biotechnology Information (NCBI) database (Appendix A). The real-time qPCR, used to monitor expression changes upon treatment, was performed using the Fluidigm BioMark HD System (San Francisco, CA, USA) and TaqMan (Applied Biosystems, Waltham, MA, USA) gene expression assays, according to the manufacturer’s instructions. The analysis was performed using the Gentyane platform (INRAE, Clermont-Ferrand, France). Results were analysed using the Fluidigm software and expressed after normalization relative to the housekeeping gene expression. The list of primers used is provided in Appendix A with the reference genes shaded in grey.

### 2.10. Statistical Analysis

All numeric data were derived from at least three independent experiments and are shown as mean ± standard deviation. Statistical analyses were carried out using the *t*-test and Kruskal–Wallis test. *p* < 0.05, *p* < 0.01, or *p* < 0.001 were considered significant. The FactoMineR package was used to perform the principal component analysis (PCA).

## 3. Results

### 3.1. Splicing Genes Are Associated with Specific Signature Profiles

First, we studied the expression of 12 oncogenes and tumour suppressor genes involved in splicing using a Fluidigm array (Figure 1 and Appendix A) in progenitor cells (CD34^+^CD15^−^ sorted cells) from 15 patients diagnosed with CP-CML (patient 1 to 15, Appendix A) and in similar cell subsets from 10 healthy donors (HD). Compared with the HD group, five genes were significantly deregulated in CD34^+^CD15^−^ CP-CML cells (Figure 1A): four were upregulated (*hnRNPA1*, protein arginine methyltransferase 5 (*PRMT5*), Pre-MRNA processing factor 8 (*PRPF8*), and polypyrimidine tract binding protein (*PTBP1*); *p* < 0.001) and one was downregulated (serine and arginine-rich splicing factor 1 (*SRSF1*); *p* < 0.05). The principal component analysis (PCA) allowed us to identify three patient groups with distinct splicing gene expression profiles in CD34^+^CD15^−^ CP-CML cells, suggesting various levels and/or mechanisms of deregulation (Figure 1B,C and Appendix A). In group 1, most of the studied genes (9–10 genes, 5-fold change) were upregulated; in the second group, most genes were downregulated (7–10 genes, 8.7-fold change); in the third group, the expression of most genes was not deregulated (Appendix A). Moreover, in CD34^+^CD15^−^ CP-CML cells, the expression of some genes (serine and arginine-rich splicing factor 1, 3, and 10 (*SRSF1*, *SRSF3*, *SRSF10*), U2 small nuclear RNA auxiliary factor 1 (*U2AF1*), muscleblind-like splicing regulator 3 (*MBNL3*) was strongly correlated (Appendix A). On the other hand, these three groups were not correlated with two routinely used prognostic scores (Sokal, ELTS) or with the molecular response to TKI at month 6 of treatment (Appendix A).

Next, to compare the splicing gene expression profile of immature and mature CP-CML cells, we analysed the same 12 splicing genes in immature CP-CML CD34^+^CD15^−^ cells (*n* = 15 patients) and mature CP-CML peripheral blood leukocytes (PBL; *n* = 19 patients; not paired with the other group; see Material and Methods) (patient 29 to 47, Appendix A). We found that the expression of 10 of these genes was higher in mature PBC compared to immature CD34^+^CD15^−^ CP-CML cells. Only *PRMT5* displayed a higher expression in CD34^+^CD15^−^ cells (Appendix A); *SRSF1* was comparable between cell subsets. We did not detect any mutations in genes encoding classical splicing factors (*SF3B1*, *SRSF2*, *ZRSR2*, *U2AF1*; CP-CML PBC, *n* = 10) and in the non-coding U1-snRNA 12 (CP-CML PBC, *n* = 17), unlike in other blood malignancies such as MDS where splicing gene mutations are common [4].

Together, these data suggested a deregulation of splicing genes in the CP-CML cells with subtle differences between immature and mature malignant cells.

### 3.2. TKIs Affect Splicing Gene Expression

To investigate the link between the expression of these 12 splicing genes and BCR::ABL tyrosine kinase activity, we used the K562 cell line. First, the analysis of the same 12 genes implicated in splicing showed that six were significantly deregulated in K562 cells compared with HD CD34^+^CD15^−^ cells (Appendix A). Only two of these (*PRMT5*, *PRPF8*) were deregulated in CD34^+^CD15^−^ CP-CML cells. Then, we incubated K562 cells with different concentrations of two TKIs (1.7–5 µM imatinib and 0.1–1 µM nilotinib) for 24 h. The RT-qPCR analysis showed a significant and dose-dependent decrease in the expression of 11/12 genes (2-fold change for the strongest; Figure 2A,B) in treated compared with untreated cells. This suggested that their deregulation is linked to BCR::ABL tyrosine kinase activity. We obtained similar results in primary CP-CML CD34^+^ cells (*n* = 4 patients) (Appendix A) incubated with 5 µM of nilotinib for 6 days, with five genes deregulated: PHD finger protein 5A (*PHF5A*), *PTBP1*, *SRSF1*, *SRSF3*, and *SRSF10* (Table 1).

Primary CP-CML CD34^+^ cell samples were incubated with nilotinib (5 mM) for six days (*n* = 4 samples). After treatment, samples were prepared following the standard Illumina total RNA sample protocol and sequenced on a HiSeq4000 to a depth of at least 50 million reads per sample. Only five genes were significantly downregulated after 6 days of incubation with nilotinib. Patients were 57 to 60.

### 3.3. Effects of Spliceosome-Targeting Drugs in CML Cell Lines and Primary CP-CML CD34^+^ Cells

We next investigated the functional impact of spliceosome-targeting drugs in CML. First, we assessed the cytotoxic potential of three spliceosome inhibitors (SRPIN340, TG003, and EPZ015666) in K562 cells. After 72 h of incubation, K562 cell viability was reduced by all three inhibitors (Figure 3A). Calculation of the half-maximal inhibitory concentration (IC_50_) revealed concentrations at the micromolar level (Table 2). However, apoptosis was significantly increased only in cells incubated with SRPIN340 (Figure 3B). This effect was already visible at 24 h (37% of apoptosis when using 60 µM SRPIN340 versus 21% in control cells), but was higher at 72 h of incubation (35% of apoptosis when using 60 µM SRPIN340 versus 12% in control cells). This observation might be explained by the capacity of SRPIN340 to trigger apoptosis in leukaemia cell lines [25]. The three drugs did not have any effect on the cell cycle (Figure 3C). We obtained similar results in KCL22 cells (Appendix A).

The two cell lines were incubated with increasing concentrations (0–300 μM) of SRPIN340, TG003, or EPZ015666 for 72 h. Cell viability was determined using the resazurin assay. Values are the mean ± standard deviation of three independent experiments.

Analysis of the phosphorylation status of SR proteins (important players in constitutive and alternative splicing) by flow cytometry in K562 cells showed that incubation with TG003 (51 µM for 1 h 30 min) strongly reduced their phosphorylation level (Appendix A), confirming its inhibitory effect. The mean fluorescence positive/negative ratio was 10.7 ± 4.9 in control (untreated) and 6.1 ± 3.3 in treated TG003 cells (*p* < 0.05).

We then assessed the effect of these three drugs in primary CD34^+^ CP-CML cells from three patients (patients 26, 27, and 28 in Appendix A). Cell viability (at 72 h) was reduced in cells incubated with TG003 or SRPIN340 at micromolar concentrations, but not with EPZ015666 (Figure 4A). Apoptosis in primary CD34^+^ CP-CML cells was significantly increased in cells incubated with TG003 (at 50 µM and 100 µM) compared with the control (Figure 4B and Appendix A). There was a trend towards increased apoptosis in cells incubated with SRPIN340. EPZ015666 did not show any effect on apoptosis. These effects were time and concentration dependent. Importantly, all three drugs (SRPIN340 at 40 µM, TG003 at 20 µM, and EPZ015666 at 100 µM for 72 h) had no effect on cell viability and apoptosis in PBC from healthy controls (*n* = 3) (Appendix A).

Incubation with SRPIN340 and TG003, but not with EPZ015666, led to cell cycle arrest in primary CD34^+^ CP-CML (Figure 4C), an effect which was not observed in the K562 and KCL22 cell lines. Similarly, the number of cell divisions (analysed after cell labelling with CFSE to measure cell proliferation) was decreased in CD34^+^ CP-CML cells (all three patients) incubated with SRPIN340 or TG003, but not with EPZ015666, compared with the controls (Figure 4D and Appendix A). On the basis of these results, we selected TG003 for further characterization. Using fresh CP-CML peripheral blood leukocytes collected at diagnosis (*n* = 9 patients; 48 to 56 Appendix A), we assessed the TG003 effect (20 µM for 14 days in CFC assay) on the clonogenic capacity of progenitors. TG003 significantly decreased the number of colony-forming units-granulocyte/monocyte (CFU-GM) (mean decrease: 77.5 ± 12.9% versus untreated cells; Figure 5A) in CD34^+^ CP-CML cells, but not in CD34^+^ cells from healthy controls (*n* = 3) (Figure 5B). Moreover, the TG003 effect on the CFU-GM number was stronger when CD34^+^ CP-CML cells were pre-incubated with imatinib for 48 h before the CFC assay (Figure 5C,D).

## 4. Discussion

The involvement of the spliceosome, an essential machinery for the pre-mRNA splicing regulation, in oncogenesis is a recent finding [29]. In haematology, alterations (mainly point mutations) of genes’ encoding factors that are a part of the spliceosome are particularly frequent in myeloid malignancies (MDS, acute myeloid leukaemia, chronic myelomonocytic leukaemia). They represent a new therapeutic target, and spliceosome-targeting therapies are currently being developed [30]. Few data are available on genes implicated in the splicing regulation in CML, particularly in the early chronic phase. Some authors reported alterations in genes that are part of the spliceosome complex, such as *hnRNPA1* and *SRSF1* [31,32]; however, these changes could be the result of a more global dysregulation of the spliceosome machinery [31].

Here, we analysed the expression of 12 spliceosome genes in CD34^+^ CP-CML samples at diagnosis (*n* = 15). We found that five of these genes (*hnRNPA1*, *PRMT5*, *PRPF8*, *PTBP1*, *SRSF1*) were deregulated in CD34^+^ CP-CML samples compared with healthy controls. Moreover, the expression of 10/12 genes was higher in mature CP-CML PBC than in immature CD34^+^CD15^−^ CP-CML cells, suggesting intraclonal heterogeneity. On the other hand, we did not find any mutation in *SF3B1*, *SRSF2*, *ZRSR2*, *U2AF1* and the non-coding U1-snRNA (Appendix A).

Our finding concerning *SRSF1* downregulation in CD34^+^CD15^−^ CP-CML cells appears in contradiction with the increased expression reported by Sinnakannu J.R. et al. [32]. However, we included a higher number of CP-CML samples (*n* = 15 vs. *n* = 3). Moreover, although *SRSF1* was downregulated in ten samples, it was upregulated in two samples. Therefore, the discrepancy could be explained by a selection bias. Finally, we think that the choice of peripheral blood stem/progenitor cell donor samples as the control is more suitable than bone marrow or cord blood leukocytes when assessing gene expression changes in circulating CP-CML CD34^+^ cells. It could partially explain the differences between studies. The role of *SRSF1* as a biomarker of imatinib resistance from bone marrow analysis could not be confirmed by our analysis of expression in circulating CP34^+^ cells. Indeed, among our 15 patients, *SRSF1* was strongly upregulated (3.5-fold change) in two patients who were optimal responders at 12 months of TKI treatment. A comparative study of the PB and BM compartments would be interesting.

In agreement with a previous report [28], we confirmed the higher expression of *PRMT5* in CD34^+^CD15^−^ cells from patients with CP-CML compared with healthy controls. Moreover, *PRMT5* expression was higher in stem/progenitor cells compared to total peripheral blood leukocytes, suggesting that this gene may have a major pro-tumour role in the LSC compartment. As *PRMT5* was overexpressed in all tested CD34^+^CD15^−^ CP-CML samples, it could be a candidate biomarker of CP-CML stem/progenitor cells (Figure 1).

*PRPF8*, a tumour suppressor gene in myeloid malignancies [33] was also found to be upregulated, but with a more heterogeneous profile. Indeed, in 4/12 CD34^+^CD15^−^ CP-CML samples (group 2 of the splicing signature) it was downregulated. However, we could not associate its differential deregulation with a specific clinical response.

*PTBP1* and *hnRNPA1* were also upregulated in CD34^+^ CP-CML cells. A previous study reported that *hnRNPA1* is upregulated in primary bone marrow mononuclear CML cells [34]. In agreement with this, we found that it is upregulated in immature CP-CML cells (versus HD) and even more in the total PBC from patients with CP-CML. Previously, *PTBP1* upregulation has been described only in the blast phase of CML [35]. Interestingly, *PTBP1* and *hnRNPA1* functionally cooperate and regulate the alternative splicing of pyruvate kinase M, which is associated with abnormal cell metabolism [36]. Moreover, the *PTBP1* and *hnRNPA1* expression in glioma and neuroblastoma is regulated by MYC [37,38], a protein which is often overexpressed in cancer and can impair alternative splicing [39]. MYC is an essential molecular hub in CML pathophysiology [40] and could actively participate in the dysregulation of *PTBP1* and *hnRNPA1* as well as *PRMT5* and *SRSF1*, genes deregulated by MYC in other cancers [39]. This hypothesis needs to be investigated in future studies.

In our study, we found that patients with CP-CML at diagnosis (*n* = 15) could be classified into three groups on the basis of the expression profile of the 12 genes involved in splicing. However, these three groups were correlated neither with two routinely used prognostic scores nor with the patients’ molecular response. This lack of correlation could be explained by the small sample size and the limited follow-up time.

The analysis of the same 12 genes in the K562 cell line highlighted the deregulation of 6/12 genes, but only two (*PRMT5*, *PRPF8*) were in common with primary immature CP-CML cells, underlying their potential role already in the early CP-CML phase. *MBNL3*’s strong downregulation (x217) in this cell line is in agreement with a previous report on the BP-CML phase [41]. The expression of all studied genes (except *MBNL3*) in K562 cells was sensitive to incubation with TKIs, suggesting continued BCR::ABL dependency in the BP-CML phase. Overall, these results suggest that, in the absence of spliceosome gene mutations, which is confirmed in this study, the observed expression changes are a consequence of the oncogenic mechanisms of CML and, particularly, of the aberrant TK activity of the BCR::ABL protein. However, the classification of patients with CP-CML in three groups on the basis of their spliceosome gene expression profile suggests the existence of other/additional molecular mechanisms involved in these differences.

Therefore, we studied the in vitro impact of TKIs and spliceosome-targeting molecules on CML cells. Recently, several agents which target the spliceosome have been developed [6], but none of them have been tested in primary CML cell samples. Here, we used three different drugs (SRPIN340 (SRPK inhibitor), TG003 (CLK inhibitor), and EPZ015666 (*PRMT5* inhibitor)) in CML cell lines and in primary CP-CML cell samples. In K562 and KCL22 cells, we found IC_50_ values in the micromolar range for these drugs. Although these values could be considered high, we observed a significative functional effect in the two cell lines and also in primary CP-CML CD34^+^ cells. Notably, TG003 and SRPIN340 significantly affected cell viability and the cell cycle, induced apoptosis, and inhibited proliferation. Conversely, we did not observe any effect in primary HD CD34^+^ cells. These results suggest that (i) CML cells are particularly dependent on the modified spliceosome machinery; and (ii) pharmacological inhibition of the spliceosome can selectively affect tumour cell survival, which is in line with previous findings [18,25,42]. Moreover, TG003 increased the inhibitory effect of imatinib (a TKI currently used in the clinic) on CP-CML CFU-GM formation, suggesting that this therapeutic combination could be efficient. As TG003 and SRPIN340 cannot be used in clinical studies because of their pharmacokinetic instability [24,43], other more stable compounds with the same pharmacology effect have emerged, e.g., SM08502, a CLK inhibitor currently being assessed in advanced solid tumours in a phase I clinical trial (ClinicalTrials.gov Identifier: NCT03355066).

## 5. Conclusions

In conclusion, at diagnosis, CP-CML CD34^+^ cells could be classified into three groups on the basis of the expression profile of 12 genes implicated in splicing. Moreover, the finding that co-incubation with spliceosome-targeting drugs increases the effect of the TKI imatinib in CP-CML cells, but not in normal cells, suggests that this combination could represent a new strategy to specifically target immature CML cells. Future studies should also address the influence of spliceosome-targeted drugs on immunity since, on one hand, splicing changes are involved in inflammation and innate immunity (reviewed in Yang H. et al. [29]) and, on the other hand, drugs that target the spliceosome could trigger immune responses, as demonstrated for example in triple-negative breast cancer [44]. In the case of CML, these impacts on immunity are very interesting for a disease associated with immune dysregulation [45,46,47].

## Figures and Tables

**Figure 1 cancers-14-04695-f001:**
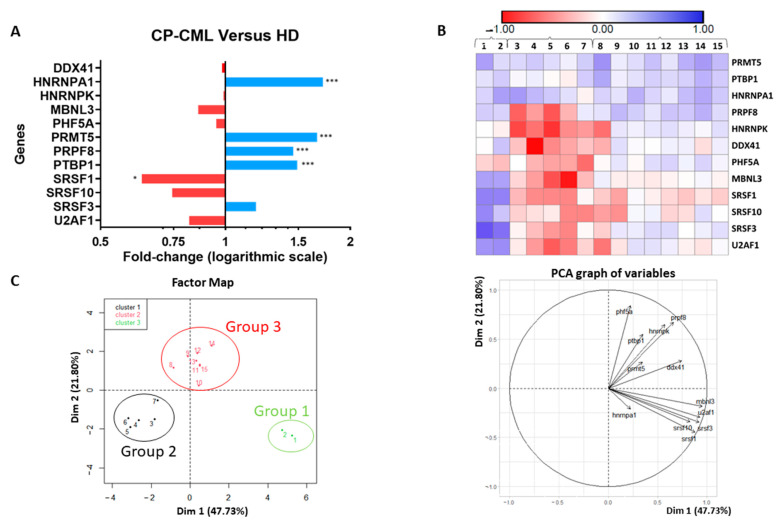
Gene expression in CP-CML CD34^+^CD15^−^ cells and HD CD34^+^CD15^−^ cells. (**A**) Gene expression was assessed with the Fluidigm BioMark HD System in CD34^+^CD15^−^ cells from patients with chronic-phase chronic myeloid leukaemia (CP-CML) (*n* = 15) and healthy donors (HD) (*n* = 10); (**B**) heatmap indicating the fold change (log2) of each CD34^+^CD15^−^ cell sample from the 15 patients with CP-CML compared with the HD samples; (**C**) PCA analysis of gene expression defining three groups of patients; * *p* < 0.05, *** *p* < 0.001 (Kruskal–Wallis test).

**Figure 2 cancers-14-04695-f002:**
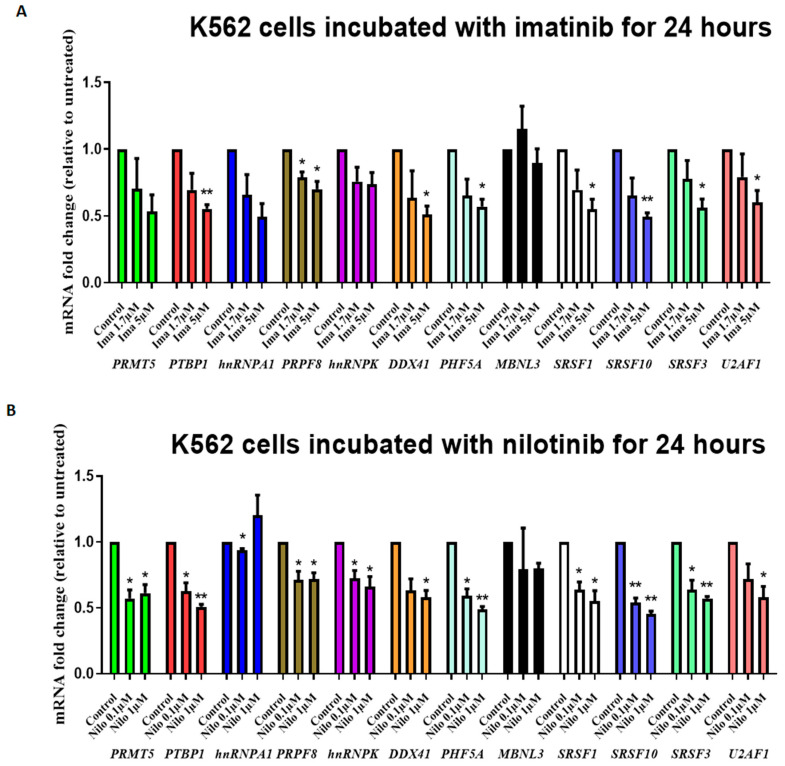
Influence of TKIs on spliceosome gene expression (**A**,**B**). K562 cells were incubated with imatinib (1.7 µM and 5 µM) or nilotinib (0.1 µM and 1 µM) for 24 h. Each color represents one gene. Gene expression is shown as fold change relative to untreated cells (control). Values are the mean ± SD of three independent experiments (*n* = 3). * *p* < 0,05, ** *p* < 0.01, (*t*-test paired).

**Figure 3 cancers-14-04695-f003:**
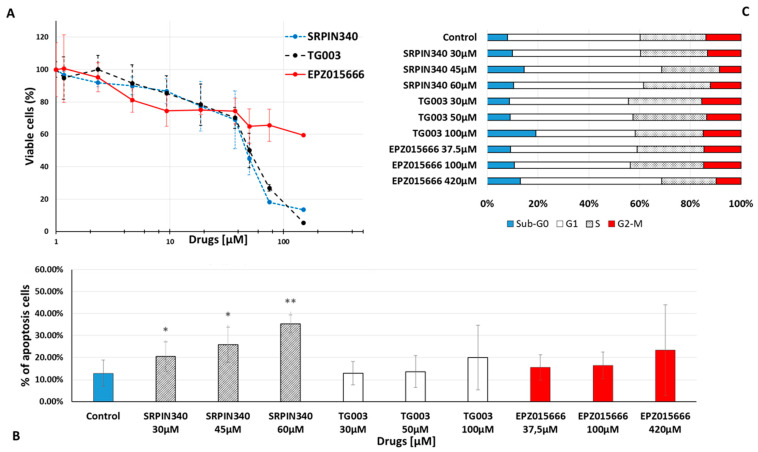
Effect of spliceosome-targeted drugs on K562 cell viability, death, and cell cycle. (**A**) K562 cells were incubated with increasing concentrations (0–300 μM) of SRPIN340, TG003, or EPZ015666 for 72 h. Cell viability was determined using the resazurin assay. Viability of cells incubated with vehicle (control) was set at 100%. Viability in treated cells was calculated as the percentage relative to control. Values are the mean ± standard deviation of three independent experiments (*n* = 3). (**B**) To assess cell death, K562 cells were incubated with three concentrations of each drug for 72 h. Control: cells incubated with vehicle. Then, cell death was evaluated by annexin V-FITC and 7-AAD staining. Values are the mean ± standard deviation of four independent experiments (*n* = 4). (**C**) To assess the effect of each drug on cell cycle, K562 cells were incubated with different concentrations of each drug for 24 h. Control: cells incubated with vehicle. The effect on cell cycle was evaluated by propidium iodide staining. Values are the mean of three independent experiments (*n* = 3). * *p* < 0.05; ** *p* < 0.01 (*t*-test for paired samples).

**Figure 4 cancers-14-04695-f004:**
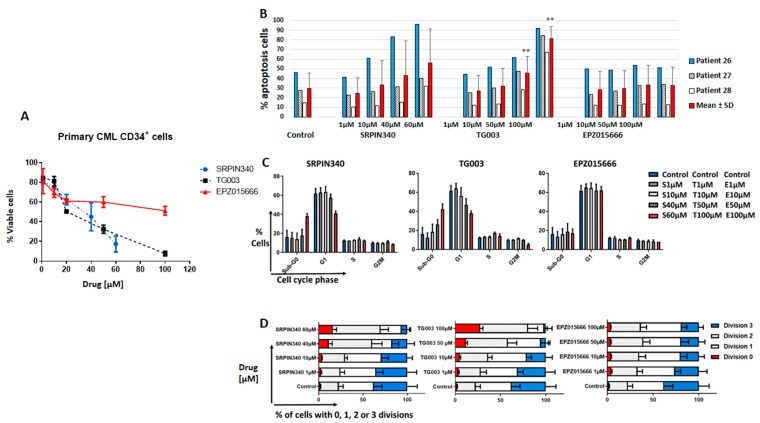
Effects of spliceosome-targeted drugs in primary CP-CML CD34^+^ cells. (**A**) Cell viability—the percentage of viable CD34^+^ CP-CML cells (*n* = 3 patients) after 72 h in culture with increasing concentrations of the indicated drugs was calculated using the Trypan Blue dye exclusion method and was relative to control (vehicle). Minimum four data points per curve (mean ± SD of three patients). (**B**) Apoptosis—primary CD34^+^ CP-CML cells (*n* = 3) were incubated with the three drugs at different concentrations for 72 h and then were stained with Annexin V and 7-AAD to assess apoptosis by flow cytometry. Values are the mean ± standard deviation of data collected from three patients. (**C**) Cell cycle—primary CD34^+^ CP-CML cells (*n* = 2) were incubated with the three drugs at different concentrations for 72 h and then were stained with propidium iodide. Values are the mean ± standard deviation of data collected from two patients. (**D**) Cell proliferation—percentage of cells with the indicated number of divisions is represented. Cell division tracking was performed using the vital fluorescent stain CFSE for 72 h. CML-CP cells were stained with the anti-CD34-APC antibody and cell division was assessed in cells in the viable gate. Cells were cultured with or without the indicated drugs at different concentrations for 72 h. Values are the mean ± standard deviation of data collected from three patients. ** *p* < 0.01 (*t*-test for paired samples).

**Figure 5 cancers-14-04695-f005:**
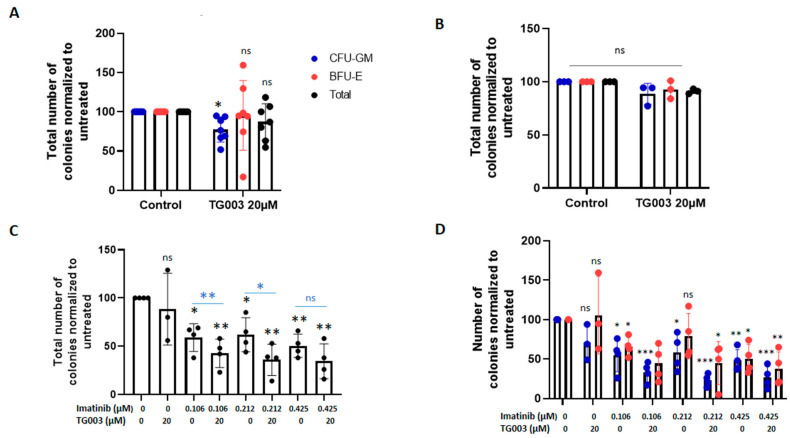
TG003 affects colony formation. (**A**) Colony formation in seven CD34^+^ samples from patients with CP-CML after incubation or not (control) with TG003. CFU-GM, colony-forming units-granulocyte/monocyte; BFU-E, burst-forming unit-erythrocyte. (**B**) Colony formation in CD34^+^ cells from healthy donors after incubation or not (control) with TG003. (**C**,**D**) Colony formation in four CD34^+^ cell samples from patients with CP-CML after incubation or not with TG003 after 48 h of pre-incubation or not with imatinib. Values are the mean ± standard deviation. * *p* < 0.05; ** *p* < 0.01; *** *p* < 0.001 (*t*-test for paired samples); patients 48 to 56 years old.

**Table 1 cancers-14-04695-t001:** TKIs affect spliceosome gene expression.

Gene	Fold Change	FDR
PHF5A	0.67	0.05
PTBP1	0.74	0.02
SRSF1	0.69	0.01
SRSF3	0.67	0.01
SRSF10	0.76	0.09

FDR: false discovery rate.

**Table 2 cancers-14-04695-t002:** Half-maximal inhibitory concentration (IC_50_) values for each drug calculated after 72 h of incubation.

Cell Line	SRPIN340	TG003	EPZ015666
K562 (CML)	45.2 ± 2.9	51.4 ± 14.4	>100
KCL22 (CML)	57.5 ± 0.6	54.0 ± 13.5	96.1 ± 0.1

## Data Availability

Not applicable.

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
