# Peer review of "The Spliceosome: A New Therapeutic Target in Chronic Myeloid Leukaemia"

_cancers, 2022, doi:10.3390/cancers14194695_

Round 1
Reviewer 1 Report
The study is well conceived and conducted. The text is complete and well written.
Suggestions about how to improve the manuscript are listed below; their locations in the text are highlighted
in the attached modified version of manuscript.
Throughout the text:
1) “peripheral blood / blood (line 376)” should become “PB”.
2) “bone marrow” should become “BM”.
Lines 24, 36 and possibly elsewhere – I would suggest to change “leukemia” to “leukaemia”, the spelling the
Authors adopted in other points of the text (together with “haematology”).
Lines 40, 83, 86, 232 - “CP-CML clone” / “CML clone” “CP-CML cell clone”: as one cannot be sure that the
CML cell population is actually monoclonal, I would suggest to shift to “CP-CML cells”, which the Authors
actually use very many times.
Lines 213-5 - change “first group” to “Group 1”, and so on, to refer more closely to what indicated in Fig. 1C.
Line 220 - change “at month 6 of TKI treatment” to “to TKI at month 6 of treatment”.
Line 224 - change “material and methods” to “Materials and Methods”.
Lines 261, 262, 418, 440, 443 - change “spliceosome-targeted” to “spliceosome-targeting”.
Lines 283, 323, 345 and possibly elsewhere – change “t-test paired” to “t-test for paired samples”.
Lines 314, 317 – change “in samples” to “of data collected”.
Line 322 – change “Values are expressed as the mean ± standard deviation of three patients” to “Values are
the mean ± standard deviation of data collected from three patients”.
Line 325 – “in K562 and KCL22 cell lines” should be changed to either “in the K562 and KCL22 cell lines” or “in
K562 and KCL22 cells”.
Line 387 – change “CD34+ CP-CML” to “CD34+ CP-CML cells”.
Line 391 – change “in the BP-CML phase” to either “in the blast phase of CML” or “BP-CML”.
Line 393 – change “M” to “M,”
Line 397 – change “;” to “,”.
Line 424 – change “IC50” to “values.
Line 430 – change “finding” to “findings”.
Lines 443-4 – change “on the one hand” to “, on one hand,”.
Line 445 – change “and on” to “and, on”.
Line 446 – add a comma after “responses”
Line 446 – change “triple negative” to “triple-negative”.
Author Response
Dear reviewer,
We have made all the changes requested in terms of content and form.
Kind regards,
Benjamin Lebecque
Reviewer 2 Report
Authors investigated the expression of 12 splicing genes (oncogenes and tumor suppressor genes) in chronic phase (CP) chronic myeloid leukemia (CML) CD34+ cells at diagnosis of 15 patients. They found a distinct splicing signature profile in comparison with healthy donor CD34+ cells (10 healthy donors). Results showed a spliceosome deregulation and an intra-clonal heterogeneity in CML. Two spliceosome inhibitors (TG003 and SRPIN340) significantly affected cell viability and cell cycle, induced apoptosis, and inhibited proliferation in KCL22 and K562 cell lines and in primary CP-CML CD34+ cells but were without effect in primary helthy donors CD34+ cells. TG003 increased the inhibitory effect of imatinib. Results are interesting and alternative splicing of pre-mRNA has not been adequately studied in CML so far. Abbreviation of some genes are not cleared (hnRNPA1, PRMT5, PRPF8, PTB1, SRSF1). Three used spliceosome inhibitors should be better described.
Author Response
Dear reviewer,
We have made all the changes requested in terms of content and form.
We have added a paragraph in the material and method that specifies the characteristics of each of the three drugs used.
"TG003 is a benzothiazole compound that inhibits the kinase activity of Cdc2-like kinase (Clks). CLK1 is alternatively spliced in various cancer cell lines including K562[15]. By this intermediate, TG003 affects the phosphorylation of serine/arginine-rich protein (SR proteins) that mediate the regulation of alternative splicing in vitro and in vivo and blocks dissociation of nuclear speckles16. This drug has already different applications; TG003 was used in cancer models [17,18], in genetic disease such as Duchenne muscular dystrophy[19] or in influenza A virus replication [20].
SRPIN340 is an ATP-competitive inhibitor of Serine/arginine-rich protein kinases 1 and 2 (SRPKs) [21]. SRPKs are known to be upregulated in CML22. This compound affects phosphorylation of SR family members (SRSF1, SRSF2, SRSF4, SRSF5 and SRSF6) and modulate the expression of genes involved in cell proliferation and survival (MAP2K1, MAP2K2, VEGF and FAS). It has been used as an antiviral agent [23] or as an anti-tumor [24–26].
EPZ015666 is a PRMT5 inhibitor. This drug has been used in different cancer as lymphoma27. PRMT5 is a key regulator of the core splicing machinery, and is known to be upregulated in CML[28]."
Kind regards,
Benjamin Lebecque